# Self-training with Selective Re-training Improves Abdominal Organ Segmentation in CT Image

Fan Zhang[1], Meihuan Wang[2], and Hua Yang[1]([✉])

[1] Department of Radiological Algorithm, Fosun Aitrox Information Technology Co., LTD., Shanghai, China.
[2] College of Medicine and Biological Information Engineering, Northeastern University, Shenyang, China
{yanghua}@fosun.com

**Abstract.** Inspired by self-training learning via pseudo labeling, we construct self-training framework with selective re-training pseudo labels to improve semi-supervised abdominal organ segmentation. In this work, we carefully design the strong data augmentations (SDA) and test-time augmentations (TTA) to alleviate overfitting noisy labels as well as decouple similar predictions between the teacher and student models. For efficient segmentation learning (ESL), knowledge distillation is adopted to transfer larger teacher model to smaller student model for compressing model. In addition, we propose the single-label based connected component labelling (CCL) for post processing. Compared to one-hot CCL of O(n) time complexity, which on the single-label based method is reduced to O(1). Quantitative evaluation on the FLARE2022 validation cases, this method achieves the average dice similarity coefficient (DSC) of 0.8813 on semi-supervised model, it achieves significant improvement compared to 0.7711 on full-supervised model. Code is available at
https://github.com/Shanghai-Aitrox-Technology/EfficientSegLearning

**Keywords:** Self-training · Efficient segmentation learning · Abdominal organ segmentation.

## 1 Introduction

Automatic segmentation of abdominal organs is confronted with main difficulties stem from three aspects: 1) It is costly, laborious, and even infeasible to annotate multi-organs at pixel-wise level in a large dataset. 2) The limited consumption resource and segmentation efficiency are required. 3) The variations in size, morphology and texture of different organs lead to class imbalance problem.

To avert the labor-intensive procedure for voxel-wise manual labeling, semi-supervised semantic segmentation has been proposed to learn a model from a handful of labeled images along with abundant unlabeled images. The self-training is commonly regarded as a form of entropy minimization in semi-supervised learning (SSL), since the re-trained student is supervised with pseudo

labels produced by the teacher which is trained on labeled data. However, potential performance degradation when iteratively optimizing the model with those ill-posed pseudo labels.

For efficient segmentation learning (ESL), self-training and self-supervised as the label-efficient approaches are used to boost the model representation capacity. Moreover, the common model compression and acceleration methods including pruning, distillation and quantization are adopted to produce light-weight models for efficient inference. The main concern of these method is to avoid the potential performance degradation on compressed model.

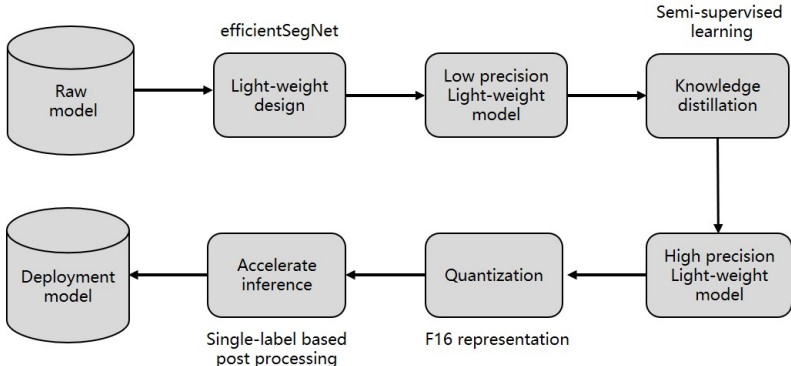

**Fig. 1.** A schematic diagram of the proposed efficient segmentation learning framework.

In this work, we empirically present four simple and effective techniques to alleviate the potential performance degradation as follows:

- We adopt an advanced self-training framework performs selective re-training via prioritizing reliable images based on holistic prediction-level stability in the entire training course.
- We design strong data augmentations (SDA) and test-time augmentations (TTA) on unlabeled images to alleviate overfitting noisy labels as well as decouple similar predictions between the teacher and student.
- We adopt knowledge distillation to transfer the knowledge from larger teacher model to smaller student model.
- We convert the one-hot based connected component labelling (CCL) to single-label based CCL for post processing.

## 2   Method

The pipeline of the proposed efficient segmentation framework is depicted in Fig 1. We adopt the whole-volume-based coarse-to-fine framework as proposed in efficientSegNet [8] for abdominal multi-organ segmentation. The self-training

is adopted for semi-supervised semantic segmentation. In addition, post quantization and single-label based CCL are designed to accelerate the inference. A detail description of the method is as follows.

### 2.1 Preprocessing

The proposed method includes the following preprocessing steps:

- Reorienting images to the right-anterior-inferior (RAI) view by linear resampling.
- Background removal by threshold segmentation. Cropping the bounding box of target, and resampling image to fixed size. The sizes of coarse and fine input are [160, 160, 160] and [160, 192, 192], respectively.
- Intensity normalization: First, the image is clipped to the range [-500, 500]. Then a z-score normalization is applied based on the mean and standard deviation of the intensity values.

### 2.2 Proposed Method

The proposed method is derived from self-training framework [7] (namely ST++) in semi-supervised semantic segmentation task. We employ 3D UNet with residual block (ResUNet) for both teacher and student models. The self-training from is as follows:

**1) Strong data augmentations**

**Table 1.** The framework of self-training with selective re-training.

| |
|---|
| **Step 1. Supervised Learning:** Train a teacher model T with higher input resolution on labeled image with weak data augmentation. |
| **Step 2. Pseudo Labeling:** Predict un-labeled image on three model checkpoints with T to obtain reliable scores. Select R highest scored images to generate pseudo labels with test-time-augmentation. |
| **Step 3. Re-training:** Re-train a student model S with equal or larger model on the jointed labeled and 50% of highest reliable pseudo labels. Where the labeled image in training phase with weak data augmentation, while pseudo labels with strong data augmentation. |
| **Step 4. Re-labelling:** Putting back the S as a T to obtain pseudo labels on un-labelled image. |
| **Step 5. Re-training:** Re-train a student model S on the jointed labeled and all of pseudo labels which reliable scores exceed the 0.9. |
| **Step 6. Update:** Return to step. 4 and employ the S model as the T model until reaching desired number of iterations. |

The weak or basic augmentations adopted in regular fully-supervised semantic segmentation, including random rotating, resizing, brightness, cropping and flipping. We inject SDA on unlabeled images to alleviate overfitting noisy labels

as well as decouple similar predictions between the teacher and student, including color, noise and painting jitter. In the pseudo labeling phase, all unlabeled images are predicted with test-time augmentations, which contains rotating, cropping and fliping.

**2) Selective re-training**

We adopt a selective re-training scheme via prioritizing reliable unlabeled samples to safely exploit the whole unlabeled set in an easy-to-hard curriculum learning manner. The measurement for the reliability or uncertainty of an unlabeled image is to compute the holistic stability of the evolving pseudo masks in different iterations during the entire training course. Therefore, the more reliable and better predicted unlabeled images can be selected based on their evolving stability during training.

Concretely, several model checkpoints are saved in the first stage supervised training, and the discrepancy of their predictions on the unlabeled image serves as a measurement for reliability. Since training model tends to converge and achieve the best performance in the late training stage, we evaluate the mean Dice between each earlier pseudo mask and the final mask. Obtaining the stability score of all unlabeled images, we sort the whole unlabeled set based on these scores, and select the top R images with the highest scores for the first retraining phase.

**3) Knowledge distillation**

The teacher model has higher input resolution and wider initial channels by giving the teacher model enough capacity and difficult environments in terms of noise to learn through. In the last iteration phase, we train a small and fast student model for inference via knowledge distillation.

### 2.3   Post-processing

We convert full precision to half precision models on the inference phase. The CCL is applied on the coarse and fine model output to remove outlier and isolated objects. The one-hot labels are converted into single-label mask, and small isolated object removal is performed on the single-label mask. Compared to O(n) time complexity of one-hot processing, this method reduces the time complexity to O(1).

## 3   Experiments

### 3.1   Dataset and evaluation measures

The FLARE 2022 is an extension of the FLARE 2021 [4] with more segmentation targets and more diverse abdomen CT scans. The dataset is curated from more than 20 medical groups under the license permission, including MSD [6], KiTS [2,3], AbdomenCT-1K [5], and TCIA [1].

The training set includes 50 labelled CT scans with pancreas disease and 2000 unlabelled CT scans with liver, kidney, spleen, or pancreas diseases. The

validation set includes 50 CT scans with liver, kidney, spleen, or pancreas diseases. The testing set includes 200 CT scans where 100 cases has liver, kidney, spleen, or pancreas diseases and the other 100 cases has uterine corpus endometrial, urothelial bladder, stomach, sarcomas, or ovarian diseases. All the CT scans only have image information and the center information is not available.

The evaluation measures consist of two accuracy measures: Dice Similarity Coefficient (DSC) and Normalized Surface Dice (NSD), and three running efficiency measures: running time, area under GPU memory-time curve, and area under CPU utilization-time curve. All measures will be used to compute the ranking. Moreover, the GPU memory consumption has a 2 GB tolerance.

### 3.2 Implementation details

**1) Data augmentations**
For weak data augmentations, the training images are randomly rotating on the x-y plane, flipping along each axis, resizing scale from 0.8 to 1.2, brightness from -200 to 200 and cropping. For the SDA on the unlabeled images, we use color jitter with random brightness, contrast and gamma, noise jitter with gaussian noise and blur, image in-painting with random values filled.

**2) Test time augmentation**
In the pseudo labeling phase, all unlabeled images are predicted with TTA, which contains rotating 180 degree along the z axis and cropping with central coordinates. The single-label based CCL is adopted to remove small isolated objects and the images are evaluated on their original resolution.

**3) Selective re-training**
The reliable images are measured with three checkpoints that are evenly saved at 1/3, 2/3, 3/3 total iterations during training. We simply treat the top 50% highest scored images with meanDice score larger than 0.9 as reliable ones and the remaining ones as unreliable. We oversampling labelled image to around the same scale as un-labelled image and then sampling uniformly from the combined dataset.

**4) Environments and requirements**
The environments and requirements of the proposed method is shown in Table 2.

**5) Training procedure**
We maintain the same optimizer strategy to train the teacher and student model. Specifically, the batch size is set as 1 with single NVIDIA 2080Ti GPU on distributed training. We use the adamW optimizer for training, where the initial base learning rate is set as 0.001. We use the step scheduling at 2/3, 6/7 epochs to decay the learning rate as 1e-4 and 5e-5 during the training process. The

**Table 2.** Environments and requirements.

| | |
|---|---|
| Ubuntu version | 16.04.10 |
| CPU | Intel(R) Xeon(R) Gold 5218 CPU @ 2.30GHz (×4) |
| RAM | 502G |
| GPU | NVIDIA 2080Ti (×8) |
| CUDA version | 11.0 |
| Programming language | Python 3.6 |
| Deep learning framework | Pytorch (torch 1.5.0, torchvision 0.8.0) |
| Code is publicly available at | Shanghai-Aitrox-Technology/EfficientSegLearning |

model is trained for 1000 epochs on the labelled image, 100 epochs on the labelled and pseudo labelled image in the first iteration phase, and 60 epochs in the subsequent iteration phase. Empirically, the 5 times of iterative training could reach the satisfying result.

The training protocols of the proposed method is shown in Table 3.

**Table 3.** Training protocols.

| | |
|---|---|
| Basic network | ResUNet with initial channels of 16 |
| Network initialization | Kaiming normal initialization |
| Batch size | 8 |
| Patch size | Coarse: 160, 160, 160 
 Fine: 160, 192, 192 |
| Optimizer | Adam with betas(0.9, 0.99), L2 penalty: 0.00001 |
| Loss | Dice loss |
| Dropout rate | 0.2 |
| Initial learning rate (lr) | 0.001 |
| Learning rate decay schedule | epoch <= epochs * 0.66: initLR 
 epochs * 0.66 < epoch <= epochs * 0.86: initLR * 0.1 
 epochs * 0.86 < epoch: initLR * 0.05 |
| Training time per iteration | 20 hours |

## 4 Results and discussion

### 4.1 Quantitative results on validation set

Quantitative result is illustrated in Table 4, it can be found that the proposed method can achieve very promising results on large organs, such as the liver, spleen, kidney, stomach. But for small organs, it remains very challenging and

also desires to pay more attention, especially for some extremely small and unclear boundary organs , such as adrenal and duodenum. Compared to full-supervised model, the proposed semi-supervised method achieves the significant improvement.

**Table 4.** Quantitative results of validation set in terms of DSC.

| Organs | Full-supervised | Semi-supervised |
|---|---|---|
| Liver | 0.9198 | 0.9771 |
| RK | 0.8620 | 0.9253 |
| Spleen | 0.8777 | 0.9762 |
| Pancreas | 0.7452 | 0.8839 |
| Aorta | 0.9286 | 0.9667 |
| IVC | 08512 | 0.9172 |
| RAG | 0.6779 | 0.7791 |
| LAG | 0.5352 | 0.7415 |
| Gallbladder | 0.5769 | 0.7971 |
| Esophagus | 0.7706 | 0.8497 |
| Stomach | 0.7896 | 0.9120 |
| Duodenum | 0.6494 | 0.7954 |
| LK | 0.8397 | 0.9361 |
| **Mean** | 0.7711 | 0.8813 |

### 4.2   Qualitative results on validation set

Fig 2 presents some easy and hard examples on validation set, and quantitative result is illustrated Table 5.For Case #21 and Case #35, our method successfully identify all organs with high DSC scores.For Case #2 and Case #44,Our method also performed well on large organs with clear boundaries,such as the spleen , but performed poorly on some organs with unclear boundaries or small organs, and even failed to segment, such as the stomach in Case #2.

### 4.3   Segmentation efficiency results on validation set

The average running time is 13.0 s per case in inference phase, and average used GPU memory is 2478 MB. The area under GPU memory-time curve is 13658.8 and the area under CPU utilization-time curve is 246.8.

### 4.4   Results on final testing set

Quantitative result is illustrated in Table 6, it can be found that in the final test set, our test results are an average DSC of 0.8860 and an average NSD of

**Table 5.** The DSC scores of easy and hard examples.

| Organs | 0021 | | 0035 | | 0002 | | 0044 | |
|---|---|---|---|---|---|---|---|---|
| | **Full** | **Semi** | **Full** | **Semi** | **Full** | **Semi** | **Full** | **Semi** |
| Liver | 0.9797 | 0.9862 | 0.9757 | 0.9833 | 0.8927 | 0.9844 | 0.9175 | 0.9807 |
| RK | 0.9834 | 0.9791 | 0.9798 | 0.9833 | 0.8731 | 0.9510 | 0.0000 | 0.0000 |
| Spleen | 0.9871 | 0.9906 | 0.9885 | 0.9891 | 0.9754 | 0.9876 | 0.9650 | 0.9873 |
| Pancreas | 0.9254 | 0.9405 | 0.8618 | 0.9149 | 0.6832 | 0.8199 | 0.5878 | 0.8093 |
| Aorta | 0.9698 | 0.9721 | 0.9689 | 0.9729 | 0.9089 | 0.9574 | 0.8642 | 0.9464 |
| IVC | 0.9412 | 0.9491 | 0.9120 | 0.9368 | 0.8452 | 0.8524 | 0.8832 | 0.9449 |
| RAG | 0.8662 | 0.8525 | 0.6510 | 0.7973 | 0.5388 | 0.7670 | 0.0000 | 0.0000 |
| LAG | 0.6821 | 0.8503 | 0.6899 | 0.8464 | 0.7880 | 0.7540 | 0.5391 | 0.8424 |
| Gallbladder | 1.0000 | 1.0000 | 0.9358 | 0.9626 | 0.3066 | 0.2166 | 0.4060 | 0.9323 |
| Esophagus | 0.9073 | 0.9218 | 0.8539 | 0.9057 | 0.0000 | 0.0000 | 0.7171 | 0.8899 |
| Stomach | 0.9548 | 0.9705 | 0.8736 | 0.9669 | 0.5553 | 0.9624 | 0.4528 | 0.9168 |
| Duodenum | 0.9052 | 0.9333 | 0.8755 | 0.8910 | 0.7173 | 0.8500 | 0.0682 | 0.5606 |
| LK | 0.9830 | 0.9806 | 0.9847 | 0.9894 | 0.9344 | 0.9700 | 0.9257 | 0.9739 |
| **Mean** | **0.9296** | **0.9482** | **0.8885** | **0.9338** | **0.6938** | **0.7748** | **0.5636** | **0.7526** |

**Table 6.** Quantitative results on final testing set.

| Organs | DSC | NSD |
|---|---|---|
| Liver | 0.9786±0.0259 | 0.9859±0.0306 |
| RK | 0.9477±0.1685 | 0.9507±0.1718 |
| Spleen | 0.9517±0.1435 | 0.9560±0.1496 |
| Pancreas | 0.8536±0.0647 | 0.9488±0.0569 |
| Aorta | 0.9648±0.0227 | 0.9831±0.0294 |
| IVC | 0.9193±0.0625 | 0.9319±0.0757 |
| RAG | 0.8276±0.1130 | 0.9404±0.1118 |
| LAG | 0.8126±0.1221 | 0.9106±0.1261 |
| Gallbladder | 0.7792±0.3339 | 0.7894±0.3386 |
| Esophagus | 0.8166±0.1166 | 0.9056±0.1182 |
| Stomach | 0.9352±0.0471 | 0.9631±0.0555 |
| Duodenum | 0.7788±0.1049 | 0.9139±0.0818 |
| LK | 0.9519±0.1363 | 0.9562±0.1429 |
| **Mean** | **0.8860±0.0755** | **0.9335±0.0501** |

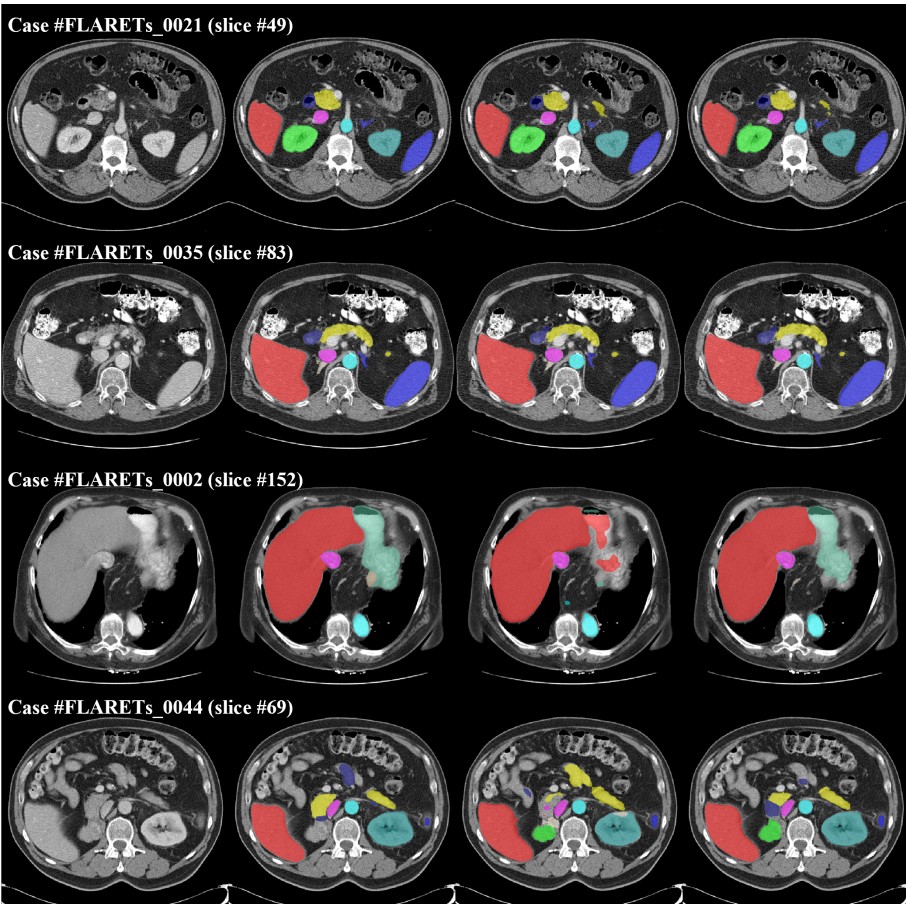

**Fig. 2.** Qualitative results of full-supervised and semi-supervised model on easy (case FLARETs 0021 and 0035) and hard(case FLARETs 0002 and 0044) examples. First column is the image, second column is the ground truth, third column is the predicted results by full-supervised model and forth column is the predicted results by semi-supervised model.

0.9335, and the average variance is very small, which proves that our model has excellent generalization.

### 4.5 Limitation and future work

More verification experiments could be performed to reduce resource consumption: 1) Lower dimension input, such as multi-views or 2.5D images. 2) Lower precision representation, such as 8 bit-widths numerical precision. 3) Training-aware pruning and quantization methods may recover the performance.

## 5 Conclusion

The proposed method achieves the highly generation ability for large organs. The main challenge in this task lies in complex anatomical structures, the unclear boundary of soft tissues, high resolution of images, and extremely unbalanced sizes among large and small organs, etc. The proposed SSL method with low-resource consumption achieves the significant improvement compared to the full-supervised method.

**Acknowledgements** We sincerely appreciate the organizers with the donation of FLARE2022 dataset. The authors of this paper declare that the segmentation method they implemented for participation in the FLARE 2022 challenge has not used any pre-trained models nor additional datasets other than those provided by the organizers. The proposed solution is fully automatic without any manual intervention.

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
