# OpenReview forum: "Self-training with Selective Re-training Improves Abdominal Organ Segmentation in CT Image"
_MICCAI.org/2022/Challenge/FLARE_

### Official Review · Reviewer_oRh5 · 2022-09-14
**improve the wording.**

**Rating:** 9
**Confidence:** 4

**Review:**

Comments to the Author

Based on previous work, the author construct self-training framework with selective re-training pseudo labels to improve semi-supervised abdominal organ segmentation, and achieves good accuracy and inference speed.

- The resolution of two images in the article is insufficient.
- Should the "Selective re-training" strategy make the model prefer CT images that are easy to be segmtioned, thereby weakening the ability to generalize? If not, why not?
- Please explain why the three checkpoints were chosen evenly throughout the iteration so that the reader can understand.
- Table 3 should show the total training time instead of the training time per iteration.
- In Section 4.4, "This is a placeholder. We will send you testing Results after the challenge." should not exist.
- The text of this article needs further polishing.


Please go through the paper and improve the wording.

---

### Official Review · Reviewer_nB1y · 2022-09-16
**Self-training for semi-supervised multi-organ segmentation**

**Rating:** 7
**Confidence:** 4

**Review:**

The authors propose a self-learning framework with iterative selective retraining. A small and fast student model for inference via knowledge distillation is trained in the last training iteration phase to decrease resource consumption.

Pros:
. Well-written paper, methods are thoroughly described.
. The method achieves good performance on the validation set.
. The proposed semi-supervised approach can leverage unlabeled data to increase performance.

Cons:
. No ablation study is performed to assess the impact of the different components in the method, namely the individual roles of SDA, TTA, decoupling, half-precision conversion, smaller student model architecture, selective vs unselective retraining, and post-processing. With the presented results, it's impossible to say how much each component affects performance and resource consumption.

---

### Official Review · Reviewer_eahC · 2022-09-17
**This work contributes enough but lacks readability and does not perform ablation experiments**

**Rating:** 7
**Confidence:** 4

**Review:**

**Summary:**

This work employs 3D ResUNet and Teacher-Student model for semi-supervised segmentation task, proposes a selective re-training method for training teacher and student models iteratively, and improves CCL with O(1) time complexity in the inference stage to improve efficiency. This work contributes enough but lacks readability and does not perform ablation experiments for the proposed method.

**Strengths:**

- Selective re-training scheme for training teacher-student model: It benefits the model to exploit unlabeled data from easy to hard.
- CCL with O(1) time complexity. It will significantly improve post-processing efficiency (if performance is as good as baseline).

**Suggest improvements:**

- No figures in this paper. If the figures replace the description in Table 1, the framework will be more readable.

- An ablation experiment may be required. As discussed in the 'Post-processing' section:

    > The one-hot labels are converted into single-label

    The proposed post-processing method is faster, but compared with the baseline method, which one is better on performance?

---

### Official Review · Reviewer_nobL · 2022-09-20
**Self-training with Selective Re-training**

**Rating:** 10
**Confidence:** 4

**Review:**

The quality, clarity, and description of the paper are good except for some minor issues.

* There is no qualitative analysis in section 4.2 (Qualitative results on validation set).
* The format of the table is different.

---

### Official Review · Reviewer_xcbg · 2022-09-20
**Good work, high quality of resulting model, proposed techniques are relevant to the task**

**Rating:** 10
**Confidence:** 5

**Review:**

Pros:
- high dice
- proposed techniques are relevant to the task
- overall, authors did a great job

Cons:
- techniques are hardly novel, yet effective

---

### Official Review · Reviewer_M7SX · 2022-09-20
**Self-training with Selective Re-training Improves Abdominal Organ Segmentation in CT Image**

**Rating:** 7
**Confidence:** 3

**Review:**

Strengths: The proposed method achieved efficient and effective semi-supervised learning with a mean DSC of 0.8813 and a mean inference time of 13 s. The proposed method used selective re-training to an effective use of the unlabeled data. Post-processing using single-label-based connected component labelling contributed to efficiency.

Weaknesses:
It would be better if the authors had suggested possible reasons for the poor segmentation results for the hard cases.

---

### Official Review · Reviewer_8v2s · 2022-09-22
**Self-training with Selective Re-training Improves Abdominal Organ Segmentation in CT Image**

**Rating:** 9
**Confidence:** 3

**Review:**

The design of the preprocessing and retraining/KD methods are very well, which makes the performance is satisfying. If more discussion about the cases is added, the well-written paper will become better.

---

### Public Comment · ~Zhengshan_Huang1 · 2022-09-21
**The article is clearly structured and contains all the required requirements**

The article is clearly structured and contains all the required requirements.

---

### Meta-Review · Program_Chairs · 2022-09-28

**Recommendation:** Minor Revision
**Confidence:** 5

**Metareview:**

Nice paper. Please address the reviewers' comments in the revised manuscript.

---

> ### Author Response · Authors · 2022-10-25
> **Self-training with Selective Re-training Improves Abdominal Organ Segmentation in CT Image**
>
> 1.Relevant citations have been added.
>
> 2.We have changed the CT image to proper window level (40) and width (400).
>
> 3.Adjust format of tables as well as resolution of images.
>
> 4.Add quantitative results on final testing set.